# A Review of Cobalt-Containing Nanomaterials, Carbon Nanomaterials and Their Composites in Preparation Methods and Application

**DOI:** 10.3390/nano12122042

**Published:** 2022-06-14

**Authors:** Hongfeng Chen, Wei Wang, Lin Yang, Liang Dong, Dechen Wang, Xinkai Xu, Dijia Wang, Jingchun Huang, Mengge Lv, Haiwang Wang

**Affiliations:** A Key Laboratory of Dielectric and Electrolyte Functional Material Hebei Province, Northeastern University at Qinhuangdao, Qinhuangdao 066004, China; chf1405104586@126.com (H.C.); wang15991035831@163.com (W.W.); yl20010508@yeah.net (L.Y.); dongliang@neuq.edu.cn (L.D.); aaa1291233239@163.com (D.W.); autxxk@163.com (X.X.); dijia8121@foxmail.com (D.W.); hjc13799078806@163.com (J.H.); doublelmg@outlook.com (M.L.)

**Keywords:** cobalt-containing nanomaterials, carbon nanomaterials, composite materials, anode of lithium-ion batteries

## Abstract

With the increasing demand for sustainable and green energy, electric energy storage technologies have received enough attention and extensive research. Among them, Li-ion batteries (LIBs) are widely used because of their excellent performance, but in practical applications, the electrochemical performance of electrode materials is not satisfactory. Carbon-based materials with high chemical stability, strong conductivity, high specific surface area, and good capacity retention are traditional anode materials in electrochemical energy storage devices, while cobalt-based nano-materials have been widely used in LIBs anodes because of their high theoretical specific capacity. This paper gives a systematic summary of the state of research of cobalt-containing nanomaterials, carbon nanomaterials, and their composites in LIBs anodes. Moreover, the preparation methods of electrode materials and measures to improve electrochemical performance are also summarized. The electrochemical performance of anode materials can be significantly improved by compounding carbon nanomaterials with cobalt nanomaterials. Composite materials have better electrical conductivity, as well as higher cycle ability and reversibility than single materials, and the synergistic effect between them can explain this phenomenon. In addition, the electrochemical performance of materials can be significantly improved by adjusting the microstructure of materials (especially preparing them into porous structures). Among the different microscopic morphologies of materials, porous structure can provide more positions for chimerism of lithium ions, shorten the diffusion distance between electrons and ions, and thus promote the transfer of lithium ions and the diffusion of electrolytes.

## 1. Introduction

With the continuous development of the times, fossil fuels, which are non-renewable resources, are decreasing day by day, and the environmental pollution problem is becoming more and more serious. It is extremely urgent to develop green renewable energy, such as wind energy and solar energy [1]. Because these energy sources are highly dependent on the natural environment, their transfer and storage become particularly important. Therefore, researchers and the energy community are very interested in the research of more efficient energy storage devices and technologies. At present, rechargeable batteries and supercapacitors are the main chemical energy storage devices. Among them, rechargeable lithium-ion batteries with fast charging and discharging speed, high energy density, high voltage, and good safety performance have been widely used in digital products, such as mobile phones and notebook computers.

Compared with traditional materials, nanostructured electrodes have the advantages of improving highly reversible lithium insertion ability, reducing diffusion length, increasing lithiation/delithiation rates, and enhancing electrical conductivity. Carbon nanomaterials represent one of the most widely studied nanomaterials at present. Because of their novel structures, mechanical properties, and electronic double layer capacitor (EDLC) behavior, most nano-carbon-based anode materials that have been developed have remarkable lithium storage and recycling properties compared with commercial graphite. The theoretical capacity of commercial graphite is 372 mA h g^−1^, and the rate performance is very limited. In addition, carbon nanomaterials also have excellent electrical and mechanical properties, so they show great application potential in LIBs, and are considered as the most commercial anode materials in lithium-ion batteries [1,2]. With the continuous exploration of researchers, it has been found that transition metal compounds with the Faraday charge transfer process can store more energy than carbon-based materials, and the reversible capacity of transition metal oxides (MO, where M = Fe, Co, Ni or Cu) is almost three times that of graphite, forcing their consideration as a promising electrode material [3,4]. Among these transition metal compounds, cobalt nano compounds, including cobalt oxides, hydroxides, sulfides, phosphates, selenides, and other derivatives, are widely used in lithium-ion batteries because of their high theoretical specific capacitance. However, most cobalt-based electrode materials are prone to large volume fluctuation during use. Due to the absorption and release of Li^+^, the electrode materials are finally crushed and separated from the current collector, which can greatly attenuate their capacity and deteriorate their recyclability. One of the ways to solve this problem is to combine carbon nanomaterials to prepare composite materials with special structure.

Up to now, there have been many reports about cobalt-based electrode materials and carbon nanomaterials in lithium-ion battery anodes. However, most of these reports introduce them separately, and there is no overview of these two materials and their composites in lithium-ion battery anodes at present. Therefore, the recent research progress of cobalt nanomaterials, carbon nanomaterials, and their composites in lithium-ion batteries is reviewed in this paper. The working mechanism of the lithium-ion battery and the application of cobalt nano-materials, carbon nano-materials, and their composite materials in lithium-ion battery anode is discussed respectively. Finally, the challenges of electrode materials at present are discussed.

## 2. Application of Cobalt-Based Nanomaterials

With the development of economy and society, people’s demand for sustainable and renewable energy is increasing day by day. In this context, power storage technology develops rapidly [5]. LIBs are widely used in green vehicles and portable electronic devices due to their high energy density, environmental friendliness, long cycle life, and almost no memory effect. Currently, lithium-ion batteries are at the forefront of current energy storage material research.

The LIBs are mainly composed of a positive electrode, negative electrode, electrolyte, and separator. The working mechanism of LIBs is the reversible intercalation and delamination of lithium ions between the positive and negative electrodes [6], thereby forming a lithium ion concentration difference between the positive and negative electrodes, and then charging and discharging through the gain and loss of electrons. As shown in Figure 1, in LiCoO_2_-graphite Li-ion batteries, lithium ions are deintercalated from the LiCoO_2_ electrode and inserted into the negative electrode (graphite) through the electrolyte and separator, which is the charging process. At this time, Co^3+^ undergoes an oxidation reaction and becomes Co^4+^, and the lost electrons are transferred to the negative electrode by means of an external circuit. The discharge reaction and the charge reaction are reciprocal. The electrode reaction formulas of the lithium-ion battery during the charging process are as follows [7]:Positive reaction: LiCoO_2_ → Li_1−x_CoO_2_ + xLi^+^ + xe^−^(1)
Negative reaction: 6C + xLi^+^ + xe^−^ → Li_x_C_6_(2)
Total reaction: LiCoO_2_ + 6C → Li_1−x_CoO_2_ + Li_x_C_6_(3)

At present, commercial lithium-ion batteries usually use 4 V cathode materials, such as spinel LiMn_2_O_4_, olivine LiFePO_4_, and layered LiNi_1/3_Mn_1/3_Co_1/3_O_2_ or LiCoO_2_. However, their energy density is limited, and the energy density of olivine LiFePO_4_ is only 120 W h kg^−1^ [8]. In recent decades, LCO (LiCoO_2_) invented by Sony has been widely used in the battery field. However, LCO is unstable in a wide range of de-lithiated state. Overcharge reaction of lithium-ion battery will change the positive electrode structure, make the material have strong oxidizability, and easily make the solvent in electrolyte oxidize strongly. Moreover, this effect is irreversible, and the large accumulation of heat released by the reaction will cause the risk of thermal runaway. Therefore, the LCO battery is not the first choice for automobiles [9]. When the charging voltage of the LCO-based battery is higher than 4.35 V, the reaction between the electrolyte solvent and the highly oxidized LCO cathode surface will occur, resulting in the instability of the electrode structure and capacity attenuation and increasing the interface impedance. To address this problem, Liu et al. developed a doping technique to solve its long-term instability, explored the co-doping of LCO with lanthanum and aluminum, and proved that doping La and Al on cobalt-containing precursors can improve the structural stability and Li^+^ diffusion rate of LiCoO_2_. This doped LiCoO_2_ shows an extremely high capacity of 190 mA h g^−1^, achieving a capacity retention rate of 96% and significantly improving its structural stability during 50 cycles of 4.5 V cutoff voltage [10].

The commonly used electrode material on the anode side is graphite, which has a relatively low theoretical capacity (372 mA h g^−1^) for commercial LIBs [11], and also suffers from the disadvantage of flammability [12].

Due to the practical demands in life, high energy density and power density are required to perform well in energy storage devices [12]. Although LIBs have a high energy density to a certain extent, there are still some problems, such as low power density, safety problems, and material aging, etc., which will limit their practical application [13]. Therefore, scholars have extensively explored the development of new electrode materials to improve the energy density and power density of lithium-ion batteries and to improve their comprehensive electrochemical performance.

In addition, a large number of experimental studies have proved that compounding cobalt nano-compounds with other materials to construct different structures plays an important role in improving the electrochemical performance of single cobalt nano-materials and has significant advantages in electrochemical energy storage devices. For example, Aboelazm et al. prepared Co_3_O_4_ with excellent performance by electrodeposition using magnetic field effects [14]. The specific capacitance of the Co_3_O_4_ nanostructure is very high (1273 F g^−1^), which is four times that of the Co_3_O_4_ nanomaterial prepared without magnetic field effect (315 F g^−1^). Such a high charge storage is due to the higher electroactive surface area contributing to faster ion diffusion and redox reactions, thus improving the pseudo capacitance in the Co_3_O_4_ nanostructure. The stability of the Co_3_O_4_ nanostructure reaches 96% after 5000 cycles, which is due to the presence of easily diffusible ions in the layered nanostructure. Ali et al. prepared Co_3_O_4_/SiO_2_ nanocomposites following the citric acid-gel method [15]. Microscopically, Co_3_O_4_ nanoparticles were uniformly embedded in SiO_2_ matrix. The electrochemical analysis results show that the prepared Co_3_O_4_/SiO_2_ nanocomposites have good charge storage properties (1143 F g^−1^ at 2.5 mV s^−1^; 679 F g^−1^ at 1 A g^−1^), which is due to the easy infiltration of electrolyte in SiO_2_ matrix and redox reaction with Co_3_O_4_ electroactive surface. After 900 times of continuous charge and discharge, the capacitance retention rate is higher than 92%, indicating that the composite has good stability. Furthermore, Ali et al. prepared activated carbon by using the CaO impregnation method with palm kernel shell (ACPKS) as raw material, and then prepared CaO/ACPKS with highly porous honeycomb structure with egg shell waste as raw material [16]. The specific capacitance of CaO/ACPKS at 0.025 A g^−1^ is up to 222 F g^−1^, which is about three times that of ACPKS. This is because the CaO/ACPKS structure stores more charge through ion intercalation/deintercalation, and the nano aperture is beneficial to the adsorption of ions on the supercapacitor.

A large number of studies have shown that the composite of cobalt compounds and carbon nanomaterials can significantly improve the electrochemical performance of lithium-ion batteries [17,18,19,20]. For example, in the research of Minakshi et al., LiCoPO_4_/C nanocomposites were prepared by solid-state fusion [21]. The introduction of amorphous carbon coating can improve the electrical conductivity and electrochemical performance. The tests show that the initial discharge capacity of the synthesized LiCoPO_4_/C nanocomposites reaches 123 mA h g^−1^, and the capacity retention is 89% after 30 cycles. Kang et al. prepared a CoOHCl@C composite with egg-like structure, in which cobalt hydroxychloride is used as egg yolk and the outer layer is wrapped with carbon shell [22]. The synergistic effect between them increases the rate performance and cycle stability of electrode materials. After 100 cycles at 2.0 A g^−1^, the discharge capacity was 665 mA h g^−1^ and the capacity retention rate was 91.5%. Therefore, this paper will discuss the application of cobalt compounds, carbon nanomaterials, and their composites in lithium-ion batteries.

### 2.1. Co_3_O_4_

Currently, LIBs anode materials include the following categories: alloys [23,24], carbon-based materials [25,26], metal oxides [27,28], metal sulfides [29,30], and metal organic frameworks (MOFs) [31,32]. In addition, Manickam Minakshi et al. used binary transition metal oxides, phosphates, and molybdenum salts as anodes for LIBs and sodium-ion batteries (SIB) and achieved good experimental results [33,34,35].The anode materials first explored for LIBs conversion reaction are first-row transition metal oxides (TMOs), such as Fe_2_O_3_, MnO_2_, NiO, and Co_3_O_4_ [36,37], of which Co_3_O_4_ has the best performance [12].

In recent years, many scholars have studied it. For example, Dona Susan Baji et al. synthesized a new discoid Co_3_O_4_ with high porosity following a simple hydrothermal method [38]. The presence of porous structure and cavity in the structure will increase the surface area of Co_3_O_4_, which can provide more locations and channels for the rapid diffusion of lithium-ions and adapt to the volume change during the charge and discharge process, resulting in a great increase in electrochemical activities. The material has a capacity of 510.5 mA h g^−1^ after 50 charge and discharge cycles under 4.0 C conditions. Li et al. prepared 2D porous Co_3_O_4_ nanosheets by self-sacrificing template method. The two-dimensional porous structure promoted the diffusion of electrolyte and the insertion/extraction of Li^+^ during charge and discharge, thus enhancing its electrochemical performance [39]. After 100 cycles at a current density of 1 A g^−1^, the discharge capacity of the material is still 1000 mA h g^−1^, indicating that the material had a high discharge capacity and excellent cyclic stability. Zhang et al. synthesized hollow/porous Co_3_O_4_ nanospheres by a simple impregnation method using carbon spheres as sacrificial templates and a metal-organic skeleton (MOF) assisted strategy, and then annealed them in air [40]. The semicircle diameter of hollow/porous Co_3_O_4_ microspheres is much smaller than that of Co_3_O_4_ nanorods, indicating that the former has a lower contact impedance and charge transfer impedance as an anode material. The hollow porous Co_3_O_4_ microspheres provided a high reversible discharge specific capacity of about 550 mA h g^−1^ and showed excellent cycle stability and rate performance after 520 cycles at 100 mA g^−1^ current density. Li et al. successfully prepared ultrathin mesoporous Co_3_O_4_ by the calcination of precursor Co^II^Co^III^-layered double hydroxide nanosheet arrays, which has an ordered sheet nanostructure and uniform mesoporous distribution [41]. The Co_3_O_4_ nanosheet arrays with medium thickness prepared by this structure have appropriate ultrathin thickness and abundant mesopores, showing their excellent electrochemical performance for LIBs. They have a high specific charge capacity of 2019.6 mA h g^−1^ at the current density of 0.1 A g^−1^, good rate performance, and significant cycle stability (There is still a high capacitance of 1576.9 mA h g^−1^ after 80 cycles). This ultrathin nano-sheet structure makes lithium-ions have a larger specific area, shortens the transmission path of lithium-ions, and improves the storage performance of lithium-ions.

These results indicate that cobalt oxide as anode of lithium battery can improve its electrochemical performance by changing its morphology.

Controlling the synthesis temperature of cobalt oxide is the key to control its morphology. For example, the Co_3_O_4_ nanostructure presents a cubic shape and is composed of nanotubes [42]. The higher the synthesis temperature is, the faster the growth rate, and then the nanotubes are obtained. The Co_3_O_4_ nanostructured material has a high capacitance of 833 F g^−1^. The material synthesized at the optimum temperature of 18 °C has high reversibility and can adsorb and desorb ions from aqueous solution.

However, when metal oxides are used as anode materials, their capacity decays rapidly, their intrinsic conductivity is relatively low, and they will have large volume changes in the processes of lithiation and delithation, so their life cycle needs to be prolonged and their performance still needs to be further optimized, which limits their practical application in batteries [11].

### 2.2. Binary Metal Oxides

Binary metal oxide ACo_2_O_4_ (A = Ni, Fe, Mg, Zn, Mn) is a material obtained by partially replacing Co_3_O_4_ with other lower cost and environmentally friendly metals. The use of this material as the anode of lithium-ion batteries has been widely studied, and it is considered to have good development prospects [43,44,45].

For example, compared with the natural content of cobalt, manganese is more abundant, and the cost is 20 times lower than that of cobalt, which can effectively reduce the cost of anode materials. Badway and his colleagues coupled Co_3_O_4_ anode and LiCoO_2_ cathode as a lithium-ion battery, which can provide a specific capacity of 120 mA h g^−1^ [46]. However, due to the high oxidation potential of Co, the average working potential of this cell is only 2.0 V. After introducing Mn into the A sites of the Co_3_O_4_ spinel structure, the actual working voltage of the battery can be increased by reducing the working voltage of the anode [12]. The synergistic effect of this bimetal also increases the theoretical capacity of the battery on the basis of Co_3_O_4_ [47,48,49].

When MnCo_2_O_4_ works as the anode active material of LIBs, it will inevitably produce volume change, which will affect the cycle performance of the battery [50,51,52]. Therefore, the researchers also explored the different morphologies [53,54,55,56,57] and composite structures of MnCo_2_O_4_ [58].

Islama et al. used the solvothermal method to prepare porous nanosheet-assembled MnCo_2_O_4_ microspheres as shown in Figure 2 [12]. The performance in practical applications is evaluated by testing the electrochemical response of the material in a cell structure. The energy density of the LiCoPO_4_/MnCo_2_O_4_ cell reaches 415 W h kg^−1^ at high operating voltage [12].

Xu et al. prepared three-dimensional porous hydrangea-like MnCo_2_O_4_ (h-MCO) by hydrothermal calcination [58]. As shown in the Figure 3, the micro morphology of MnCo_2_O_4_ is uniform rod, while hydrangea-like MnCo_2_O_4_ is spherical with roughly the same size. Zhang et al. used this three-dimensional porous h-MCO to obtain a lithium-ion battery [54]. The electrochemical reaction involved in the first discharge process of h-MnCo_2_O_4_ electrode can be expressed by the following equations [54,59,60]:MnCo_2_O_4_ + 8Li^+^ + 8e^−^ → Mn + 2Co + 4Li_2_O(4)
Mn + Li_2_O ↔ MnO + 2Li^+^ + 2e^−^(5)
Co + Li_2_O ↔ CoO + 2Li^+^ + 2e^−^(6)
CoO + 1/3Li_2_O ↔ 1/3Co_3_O_4_ + 2/3Li^+^ + 2/3e^−^(7)
where Equation (4) represents the main reaction at the cathode, the decomposition of MnCo_2_O_4_ into Co and Mn and the production of Li_2_O, which is accompanied by the growth of solid electrolyte interface (SEI) film. Equations (5)–(7) represent the main anodic reaction process, in which Mn and Co are oxidized to MnO and CoO, respectively, while Li_2_O is decomposed to Li^+^ and Co_3_O_4_ is produced.

In 100 cycles of 0.1 A g^−1^, the reversible specific capacitance of h-MCO remains at 930 A h kg^−1^ and is higher than its theoretical capacity. As shown in Figure 4a, the reversible specific capacitance remains at 855 A h kg^−1^ in 300 cycles of 0.5 A g^−1^ and the reversible specific capacitance remains at 178 A h kg^−1^ in 600 cycles of 1 A g^−1^. The specific capacity of h-MCO remains at 566 A h kg^−1^ even at a rate of 2 A g^−1^. It can be seen that h-MCO has a good capacity retention rate. Figure 4b also shows that the cycle life of h-MCO electrode is longer than that of rod-shaped MnCo_2_O_4_ (r-MCO) electrode. Figure 4c shows that in a repetitive cycle, although the volume of the h-MCO electrode has been greatly expanded, it still retains its original hydrangea shape, indicating that it has a stable three-dimensional mesoporous hydrangea-like shape of spinel structure. The electrochemical impedance spectroscopy (EIS) of MCOs before and after cycling are determined, which shows that the h-MCO electrode had excellent electrochemical performance. Figure 4d shows that the conductive network and mesoporous structure developed by h-MCO can reduce the charge transfer resistance, and the ion diffusion resistance of the h-MCO electrode is small and the electron conductivity is better. This may be due to the larger specific surface area of this material, the porous and rough surface, and its more stable three-dimensional structure [58].

In addition, there are many reports on MnCo_2_O_4_ materials with unique structures. As shown in Figure 5a,b, Jin’s team [48] and Wu’s team [61] produced MnCo_2_O_4_ mesoporous microspheres and verified their mesoporous structure respectively, and attributed the excellent electrical properties of the material to this mesoporous structure. Li et al. prepared MnCo_2_O_4_ sub-microspheres with different hollow structures of homogeneous MnCo_2_O_4_ sub-microspheres and verified that the yolk-shell structure (Figure 5c) had the best performance, because the mesoporous structure of the yolk-shell structure could improve the stability and thus the recyclability of the electrode. The hollow structure could expand the contact area between the electrode and the electrolyte and thus improve the specific capacity of the electrode [62]. Ding et al. similarly designed yolk-shell MnCo_2_O_4_-TiO_2_ sub-microspheres (Figure 5d), with the difference that MnCo_2_O_4_ acts as the core and TiO_2_ constitutes the shell [51]. The porous yolk-shell structure slows down the volume change of the electrode material during the charging and discharging process and ensures a stable cycling performance. As in Figure 5e, Zhang et al. proposed a hierarchical self-assembled MnCo_2_O_4_ nanomembrane structure, and this material exhibited good cycling stability, especially at high current densities [54]. As shown in Figure 5f, Chen et al. embedded MnCo_2_O_4_ nanoparticles uniformly in paper-like graphene sheets, and the excellent conductivity of graphene films facilitated ion/electron transport during battery operation [50].

In conclusion, by controlling the morphology of MnCo_2_O_4_ material to form a mesoporous structure, the volume change resistance of microstructure can be improved, thus improving the cycle stability of electrode. By designing the hollow structure, the contact area between the electrode and electrolyte can be enlarged, and the specific capacitance of the electrode can be increased.

### 2.3. Application of Cobalt Oxide Composite

Studies have shown that by combining two or more transition metal oxides with other components, the resulting hybrid materials can combine the advantages of different substances to optimize the electrochemical performance of metal oxides [63,64].

Co_3_O_4_/CeO_2_ is a material with high-rate capacity and high efficiency in Co_3_O_4_ matrix composites [65,66,67,68]. Kang et al. synthesized 5Co_3_O_4_/CeO_2_ (Co_3_O_4_/CeO_2_ having the molar ratio of Co/Ce = 5:1) composites by the microwave-assisted solvothermal method (Figure 6). As shown in Figure 7, the micromorphology shows that the prepared sample is composed of irregular small pieces. The initial discharge capacity of 5Co_3_O_4_/CeO_2_ is 1090.1 mA h g^−1^, and the reversible capacity is 1131.2 mA h g^−1^ after 100 cycles at the current density of 100 mA h g^−1^.The rate data indicate that the discharge capacity of 5Co_3_O_4_/CeO_2_ reaches 835.3 mA h g^−1^ when the current density is 2000 mA g^−1^. At the same time, the heterostructure of this material can also buffer the volume expansion and contraction of lithium-ion batteries during application [11].

Zhong et al. prepared Co_3_O_4_-CoFe_2_O_4_ nanocomposites by a one-step pyrolysis method. Such nanostructures are smaller in size and have a more uniform particle distribution, resulting in better electrochemical performance of the material [69]. Ding et al. synthesized Co_3_O_4_@TiO_2_ hollow composites by using an imidazole zeolite framework-67 (ZIF-67) and thermal decomposition of the TiO_2_ coating. When this material is used as an anode for LIBs, it shows a high cycle performance of 1057 mA h g^−1^ after 100 cycles at 100 mA g^−1^ [70].

### 2.4. Transition Metal Sulfide and Its Composites

In recent years, transition metal sulfides (TMS) have been gradually used as anode materials in lithium-ion batteries due to their large theoretical capacity, good redox potential, and low cost.

Similar to transition metal oxides, TMS electrodes also experience volume change during Li^+^ insertion/extraction, which affects the cycling performance of Li-ion batteries [71]. The rational construction of nanostructured CoS_x_ can not only shorten the diffusion length of Li^+^ ions and promote ion transport dynamics, but also provide buffer space for the volume change of CoS_x_ [72,73,74].

Yang et al. prepared a human spine-like CoMoS_3_ nanostructure by a MOF-mediated approach, utilizing the synergistic effect of each component to enhance the electrochemical performance of the material [75]. Yu et al. synthesized NiCo_2_S_4_ nanotubes on the basis of nickel foam, which have excellent performance as anodes for LIBs [76].

By combining nanostructured transition metal sulfides with carbonaceous materials, the electrical conductivity can be improved, buffering the volume change, and thus prolonging the cycle life. For example, many researchers have prepared CoS_x_/C nanocomposites by introducing carbon nanotubes, graphene, etc. [77,78,79,80,81].

Wang et al. fabricated a MoS_2_/CoMo_2_S_4_/Co_3_S_4_ nanostructure supported by a graphene layer for use in the anode of LIBs [71]. The material has a rate performance of 360 mA h g^−1^ at 10 A g^−1^ and a specific capacity of 770 mA h g^−1^ at 0.2 A g^−1^. As shown in Figure 8, they first obtained the ZIF-67 framework by precipitation, then coated the framework with molybdenum disulfide, and finally attached the sample to the surface of reduced graphene oxide and annealed it. CoMo_2_S_4_ is generated due to the interdiffusion of metal sulfides during annealing. MoS_2_/CoMo_2_S_4_/Co_3_S_4_ nanostructures can improve surface area and facilitate ion transport. In addition, due to the presence of two-dimensional reduced graphene oxide, the conductive distance becomes longer, and the structure is more stable. Therefore, its rate performance, specific capacity, and stability all perform well.

Zhu et al. prepared cobalt sulfide/reduced graphene oxide (CoS_x_/RGO) nanocomposites by the hydrothermal method [82]. Comparing the micro morphology of CoS_x_ (Figure 9a) and the prepared CoS_x_/RGO (Figure 9b), it is found that the aggregation of CoS_x_ particles is obviously more significant than that of CoS_x_/RGO particles. This shows that CoS_x_/RGO as a precursor can inhibit the agglomeration of CoS_x_. As shown in Figure 10a, on the GCD curve of CoS_x_/RGO, a long plateau at about 1.08 V can be observed in the first discharge curve, which is related to the intercalation reaction between Li^+^ ions and CoS_x_. Figure 10b shows that the discharge capacity of the composite at 50 cycles is 796 mA h g^−1^. At a current density of 800 mA g^−1^, CoS_x_/RGO has a reversible capacity of 425 mA h g^−1^ (Figure 10c), which is higher than that of RGO and CoS_x_, indicating that CoS_x_/RGO has better rate performance than RGO and CoS_x_. As shown in Figure 10d, through the EIS test (The equivalent circuit model is shown in the inset of Figure 10d), it can be seen that the combination of CoS_x_ and RGO reduces the charge transfer resistance of the electrode, thereby improving the lithium-ion storage performance.

In general, the construction of nanoelectrodes can shorten the Li^+^ diffusion distance and boost the ion transport kinetics. Compared with single metal sulfides, multicomponent TMS not only have higher electrical conductivity, but also have more abundant redox reactions [71].

The rational design of nanostructured CoS_x_ can not only reduce the diffusion length of Li^+^ ions, but also provide buffer space to accommodate the volume change of CoS_x_ [72,73,74]. Yu et al. constructed NiCo_2_S_4_ nanotubes on nickel foam, resulting in high-performance anodes of LIBs [76].

In addition, besides the above-mentioned cobalt-based materials, carbon-based materials with high chemical stability, strong conductivity, high specific surface area, and good capacity retention are also traditional anode materials in electrochemical energy storage devices [83,84]. Among them, graphite is the most widely used anode material in lithium-ion batteries. However, due to its low specific capacity (372 mA h g^−1^) and slow lithium diffusion kinetics, its practical application is limited [85]. In the past 30 years, people have made great efforts to explore and develop new carbon materials with high capacity and good rate performance, such as carbon nanotubes [86,87], nanofibers [88,89], and graphene [90,91].

## 3. Application of Carbon Nanomaterials

### 3.1. Application of Carbon Nanotube Materials

Since carbon nanotubes (CNTs) were first observed in the process of carbon arc discharge in Iijima in 1991, there has been an upsurge in research concerning CNTs all over the world, and their basic properties and potential practical applications have become deeply understood. Structurally, CNTs are one-dimensional hollow cylinders with radial dimensions of nanometers and axial dimensions of microns, which are composed of one or more rolled graphene sheets. CNTs are one of the perfect anode materials for lithium batteries because of their high conductivity, high chemical stability, and strong redox ability [92]. At present, a large number of studies have reported the successful use of CNTs as anodes of lithium-ion batteries. The results show that different morphologies, structures, and synthesis methods have great influence on the electrochemical properties of CNTs.

In previous research, the morphology and structure of CNTs were changed by ball milling or etching, and the reversible ability could be effectively improved. In the research of Yang et al., the electrochemical performance of multi-walled CNTs prepared by arc method can be effectively improved by etching, and the capacity retention rate of corroded CNTs are about 100% after 30 cycles [93]. In addition, electrochemical tests also show that the reversible capacity of the treated CNTs is four times that of the original CNTs. This remarkable improvement of reversibility can be attributed to the improvement of the purity of the original CNTs and the increase of active substances in the electrodes due to acid etching. At the same time, HRTEM images (Figure 11) show that the etched CNTs have new defects along the sidewalls, which are beneficial to the insertion and extraction of lithium-ions.

Etching treatment effectively improves the recycling capacity of CNTs, while the irreversible capacity of CNTs after acid etching increases by 100 mA h g^−1^ compared with the original CNTs. This result is probably due to the formation of a large number of active functional groups which can promote electrolyte decomposition during acid corrosion and oxidation, and heat treatment can effectively alleviate this irreversible capacity increase. However, the defects produced by etching treatment are often accompanied by the decrease of Coulomb efficiency.

Besides etching, ball milling technology is also an important method to improve the electrochemical performance of CNTs. Eom et al. used stainless steel balls to ball-mill purified multi-walled CNTs in stainless steel bottles and dried them at 150 °C for 5 h [94]. The Coulomb efficiency of multi-walled CNTs increases with the increase of milling time by analyzing the charge-discharge characteristics of multi-walled CNTs. Their reversible capacity can reach 641 mA h g^−1^, while that of pure multi-walled CNTs is only 351 mA h g^−1^, and their irreversible capacity is reduced to about 1/2 of that of pure CNTs. The decrease of irreversible capacity is related to the increase of density of ball-milled multi-walled CNTs. For the increase of reversible capacity, Eom and others think that this is related to the fracture of a single carbon nanotube to form an open end and the formation of surface functional groups, which promotes the insertion of lithium-ions. Although ball milling can improve the reversibility and Coulomb efficiency of CNTs, the treated CNTs have a large voltage hysteresis problem, which Eom attributed to the formation of a large number of active surface functional groups mentioned earlier. Gao et al. studied the ball milling treatment of single-walled CNTs, and thought that because single-walled CNTs are relatively defect-free materials, it is difficult to generate a large number of functional groups on the sidewalls, so their voltage hysteresis may be related to the kinetics of lithium diffusion to the inner core of CNTs [95]. Experiments show that the voltage hysteresis can be effectively alleviated by cutting CNTs into shorter tube segments, which has also been confirmed in the research of Yang et al. [96]. By analyzing the electrochemical performance of long and short CNTs, it is found that the reversible capacity of short carbon nanotube electrode reaches 266 and 170 mA h g^−1^ at current densities of 0.2 and 0.8 mA cm^−2^, respectively, which is almost twice that of long carbon nanotube electrode. Yang et al. think that the reason why long and short CNTs show this difference is related to the different lithium storage mechanism of their electrodes. For long CNTs, lithium-ions are mainly absorbed on one side of the graphene layer and combined with functional groups on the surface of CNTs, while short CNTs are mainly intercalated between graphene layers and doped into carbon nanotube microcavities. In addition, long CNTs have greater voltage hysteresis, which may be related to the surface functional groups and surface resistance of CNTs.

In addition to the above treatment of CNTs, the morphology of CNTs can be controlled by different synthesis methods, thus changing their electrochemical properties. Among them, Bulusheva et al. successfully synthesized and separated vertically arranged carbon nanotubes (VA-CNTs) on silicon substrate by meteorological assisted catalytic chemical vapor deposition method [97]. It is reported that VA-CNTs have better electrochemical performance than disordered carbon nanotube arrays. It is speculated that the reason for this may be that the vertically arranged structure increases the available surface area of electrolyte and lithium-ions, and then increases the specific capacity of CNTs.

Although CNTs have great application potential in the direction of anode materials for lithium-ion batteries, their application is limited due to their large structural defects and serious voltage hysteresis. In addition, compared with graphite, CNTs only demonstrate obvious improvement in capacity and cycle stability. Because their electrochemical performance is closely related to their structure and morphology, it is very important to control their structure and morphology in the preparation process, and the control of experimental conditions is also a crucial factor limiting their practical application.

### 3.2. Application of Graphene Materials

Graphene is a kind of material in which carbon atoms connected by sp^2^ hybridization are closely packed into a single layer of two-dimensional honeycomb structure. Graphene has the same structural unit as the above CNTs [98], and has excellent conductivity and mechanical strength. At the same time, its specific surface area is as high as 2630 m^2^ g^−1^, and it has high thermal conductivity exceeding 3 × 10^3^ W m^−1^ K^−1^ [99,100,101]. Compared with CNTs, graphene may have lower treatment cost, higher purity, and more stable electrochemical performance, so the research on graphene in the field of energy storage has always been a hot spot in the international nanotechnology field [102,103,104,105]. The reversible capacity of the corroded CNTs are about four times that of the original CNTs (reversible capacity of 300–640 mA h g^−1^). Their irreversible capacity (225 mA h g^−1^) is higher than that of the original CNTs (125 mA h g^−1^), and the capacity retention rate after 30 cycles is almost 100%. Graphene microspheres (453 mA h g^−1^) have almost twice the specific capacity of multilayer graphene (MG, 229 mA h g^−1^), and after 100 cycles the capacity is 94% of the original. Graphene nanosheets (GNSs), as anode materials for lithium-ion batteries, will have great application potential and are expected to replace CNTs [106]. In recent years, graphene paper, porous graphene nanosheets (PGNs), and graphene microspheres have also been reported as anode materials for lithium-ion batteries [107,108]. In Guo’s research, GNSs with the disordered structure were successfully synthesized from artificial graphite by oxidation, rapid expansion, and ultrasonic treatment [109]. When the current density is 1 mA cm^−2^, the reversible capacity of GNSs electrode is 554 mA h g^−1^, which is almost twice that of its raw material, artificial graphite. The results show that GNSs has higher reversible capacity, better cycle performance, and higher rates of charge-discharge performance than artificial graphite. This improvement in electrochemical performance can be attributed to the existence of more cavities or defects in GNSs which are beneficial to lithium storage and a large number of surface functional groups. In order to further improve the electrochemical properties of GNSs, the shape and structure of graphene nanosheets have become one of the important means. In one example, using copper nitrite as template and phenolic resin as a carbon source for the first time, PGNs with onion-like nano-pore junctions were successfully synthesized by graphitization at 2800 °C, which alleviated the problems of environmental protection in the preparation of traditional PGNs [110]. Different from the traditional graphene electrode, the PGNs have a relatively stable voltage platform at 0.01–0.2 V, and the specific capacity reaches 245 mA h g^−1^. The onion-like porous structure increases the contact area between electrode and electrolyte, and this in turn increases the chance of lithium-ion insertion, facilitates the rapid charge-discharge process, and improves the diffusion rate, so that it has excellent high-speed performance.

Recently, Rish et al. successfully synthesized three-dimensional multilayer graphene nanostructures (3DMGs) by catalytic graphitization and chemical oxidation-thermal reduction with biomass-based activated carbon as precursor [111]. The synthesis process is shown in Figure 12.

As an anode material of the lithium-ion battery, the reversible capacity of synthesized 3DMGs can reach 1513.2 mA h g^−1^ at 100 mA g^−1^, second only to porous graphene, and the charge retention rate is 153.7%. After 1100 cycles at a high rate of 5000 mA g^−1^, the specific capacity is 260.3 mA h g^−1^, and the charge retention rate is 114.47%, which shows good cycle stability. This excellent electrochemical ability can be attributed to the interconnected pore structure and high specific surface area of 3DMGs. This unique structure provides more active sites and a shorter lithium-ion diffusion pathway for lithium-ion diffusion, which is convenient for rapid ion transport. In addition, the gap between nanostructures can be used as a buffer layer for volume expansion during the lithium-ion process, which can effectively prevent volume damage during charge and discharge and is conducive to improving its cycle stability. Compared with the traditional method of synthesizing defective graphene, the above method uses green renewable carbon as precursor, and the preparation method is environmentally friendly, simple, and practical.

Compared with graphite, graphene has better electrochemical performance in lithium-ion battery anode materials. Because of its unique two-dimensional scalability, graphene can easily attach active material particles to the surface of other materials and is an excellent conductive carrier of battery materials. It is often polymerized with phosphate [112,113,114,115], silicate [116], and other polyanions as lithium-ion battery materials, showing more excellent electrochemical performance.

### 3.3. Application of Carbon Microspheres

In recent years, micron-sized carbon spheres (CSs) have been widely used in energy storage devices, such as LIBs [117], supercapacitors [118], lithium-sulfur batteries [119], and sodium ion batteries [120]. The research situation in recent years is shown in the Figure 13. Because of its high structural stability, high tap density, low surface volume ratio, and low viscous effect, it has great potential as anode material for LIBs [121,122,123,124]. In recent years, porous carbon materials have been studied as electrode materials. The micropores of carbon can effectively increase the specific surface area and enhance the contact between electrode and electrolyte. At the same time, the mesoporous structure can reduce the diffusion distance, thus improving the penetration and ion diffusion of electrolytes in carbon materials and realizing low resistance [125,126]. Therefore, nanoporous CSs have important research significance in the field of LIBs.

In Wang et al.’s research, layered porous carbon microspheres (HPCMs) with different surface morphology and internal structure were synthesized by alcohol emulsion technology with thermoplastic phenolic resin (PF) as carbon source and copper nitrate (CN) as template [127]. The results show that the content of copper nitrate can significantly affect the morphology and structure of samples, and then allow to prepare samples with different electrochemical properties. When the mass ratio of PF to CN is 1:4, the prepared HPCM not only keeps the structure of layered porous microspheres, but also has good electrochemical performance. At the current density of 50 mA g^−1^, it has a reversible capacity of 585 mA h g^−1^ and remains at 480 mA h g^−1^ after 70 cycles.

In industrial production, the synthesis process of nano-porous carbon spheres usually needs pressurized containers and volatile organic compounds [128], which limits its large-scale application. Shi et al. successfully synthesized HPCMs with a loose porous structure by a simple impregnation-carbonization process with MgAl-layered double hydroxides precursor as template and MgAl-layered double oxides microspheres as carbon source [129]. The spherical structure effectively shortens the diffusion distance of lithium ions and provides higher tap density, while the existence of mesopores further shortens the diffusion channel. The results of electrochemical performance analysis show that when the template/carbon source mass ratio is 1, the capacities of the samples obtained at current densities of 0.05, 0.2, and 1 A g^−1^ are 1140.5, 650.3, and 347.9 mA g^−1^, respectively.

Although the porous structure increases the specific surface area, its initial Coulomb efficiency is low because of the larger surface area and more active sites in direct contact with electrolyte. Therefore, how to avoid the porous structure facing electrolyte directly becomes the key to solve this problem. At the same time, the materials with pure porous structure have poor mechanical properties, which leads to poor cycle performance when they are used as electrode materials. Zhang et al. synthesized bivalve pomegranate-like porous carbon microspheres (PCMs) considering this problem [130]. The porous carbon frame is wrapped by a carbon coating on its surface. The microscopic image shows that the outer carbon coating has a multilayer structure, an interconnected porous network structure is formed inside, and the hollow carbon spheres are connected with each other.

This double carbon shell structure can limit the formation of most solid electrolyte membranes to the outer surface, effectively prevent electrolytes from entering the inner surface, improve its conductivity, and enhance its mechanical properties. In addition, the interconnected porous carbon framework can provide more positions for lithium ion insertion, shorten the diffusion distance between electrons and ions, and promote the transfer of lithium ions and electrolyte diffusion. Through the analysis of electrochemical test results, PCMs show an initial Coulomb efficiency of over 91%, which is much higher than other porous carbon anode materials (less than 70%). In addition, PCMs have capacity of 580 and 520 mA h g^−1^ at current densities of 1000 and 2000 mA g^−1^, respectively. The synthetic method can effectively enhance its mechanical properties by introducing carbon coating, and at the same time prevent electrolytes from entering, which effectively improves the shortcomings of poor mechanical properties and low initial Coulomb efficiency of porous nano-carbon materials.

Recently, Zhang and his colleagues efficiently converted biomass carbon into high value-added graphite by electrochemical graphitization of glucose carbon [131]. Glucose was first converted into amorphous nanospheres or microspheres carbon. Then, the amorphous carbon was graphitized electrochemically in molten calcium chloride, and the hollow microspheres graphite with nano-flake porous shell were successfully prepared. The morphology evolution of graphite was analyzed by SEM and TEM. When the amorphous microspheres carbon (AMC) was simply immersed in the 820 °C melt for 2 h (without cathodic polarization), it can be seen that amorphous carbon microspheres have no morphological change (Figure 14b). After polarization at 0.45 V for 0.25 h, some protrusions appear on the surface of carbon microspheres (Figure 14c), and AMC spheres transform into porous structures composed of nano-sheets after polarization for 2 h (Figure 14d). It can be observed from Figure 14e,f that the original amorphous carbon spheres are dense while the graphitized carbon spheres are hollow. This unique hollow and porous shell structure provides more electrolyte channels and accelerates ion transmission in the electrode. When it is used as the anode material of a lithium-ion battery, it can provide a reversible capacity of about 280 mA h g^−1^ at the charge and discharge rate of 5 C (1.86 A g^−1^). Zhang put forward the corresponding concept of “electrochemical graphitization” in research that found that cathodic polarization in molten chloride can graphitize amorphous carbon at about 820 °C, and this method has been proved to be suitable for all amorphous carbon, providing a basis for preparing sustainable graphite as a high-performance anode [132,133,134,135,136,137,138].

To sum up, carbon nanotubes, graphene and carbon microspheres have great application potential in lithium-ion batteries, but they also have certain advantages and disadvantages. Therefore, we made a simple comparison of the electrochemical properties of the three (Table 1).

As shown in Table 1, carbon nanomaterials with different morphologies show different electrochemical properties, and the influence of Co_3_O_4_ morphology on properties and control methods are also mentioned in the previous article, so controlling the morphology of carbon nanomaterials is also an extremely important method to improve electrochemical properties. In the research of Sun-Hwa Yeon et al., the morphology of graphene was successfully changed by ultrasonic treatment [141]. After ultrasonic treatment, the central folds of graphene flatten and transform into a honeycomb lattice, and the thickness of lamellae is relatively thinner. After 50 cycles at the same current density, the capacity retention of graphene before treatment is only 74%, while that of graphene nanosheets after treatment is 93%, which obviously improves the cycle performance. In addition, Sun et al. proposed to transform graphene oxide into graphene aerogel (GA) with porous three-dimensional structure by a simple hydrothermal method and freeze-drying process [142]. At the same time, it is found that the surface defects of GA can be regulated by controlling the reaction time, and the formation of these surface defects can increase the active sites. In addition, by observing the micro-morphology of graphene nanosheets, it is found that the porous structures in graphene nanosheets form an interconnected network structure. A porous structure increases the contact area between electrolyte and electrode, which can effectively transfer electrons and ions into electrodes. At the same time, the formation of defects also greatly improves the electrochemical stability and rate performance of electrodes. In addition, Lee et al. controlled the morphology of three-dimensional CNTs grown on Cu by changing the temperature [142]. The results show that the average diameter of CNTs grown at high temperature (750 °C) is 200–300 nm, while the diameter of CNTs grown at low temperature (700 °C) is only about 20 nm. Among them, the performance of short diameter CNTs is obviously better than that of long diameter CNTs, which is largely related to the diffusion distance of lithium ions. The diameter is short and the diffusion distance is short, so the lithium ion diffuses rapidly. The above results show that the electrochemical performance of the materials can be improved by changing the morphology of the materials under the control of the experimental conditions.

## 4. Application of Cobalt-Carbon Composites 

As mentioned earlier, cobalt and carbon materials have great application potential in lithium-ion battery anodes, and compounding various materials is also the main method to improve the electrochemical performance of cobalt nanomaterials. Among them, carbonaceous materials have become one of the ideal materials for composites with cobalt nanomaterials because of their high conductivity, and they have attracted wide attention in recent years. Among these materials, the more common one is to composite nano cobalt oxide and cobalt sulfide with carbonaceous materials.

### 4.1. Cobalt Oxide-Carbon Composites

As previously mentioned, pure Co_3_O_4_ with high theoretical capacity of 890 mA h g^−1^ has great application potential in the anode field of lithium-ion batteries. However, its low conductivity and large volume change during the lithiation and delithiation process leads to poor cycle performance and rate ability, which limits its commercial application. Researchers have tried to mitigate the influence of large volume changes through a large number of studies, including reducing the size of Co_3_O_4_ particles to nanoscale [143,144], morphological specialization [145,146], and combining with carbon materials [147,148] or metal oxides. Among these methods, the combination of Co_3_O_4_ and carbon material is the most promising, because carbon coating not only buffers the volume change, but also promotes the electronic conductivity. A large number of researchers have introduced different carbonaceous materials, including graphene [149,150,151], porous carbon [152], and carbon nanotubes [153,154], to improve their electrical conductivity.

Yan et al. synthesized carbon-doped Co_3_O_4_ hollow nanomaterials by an electrospinning and hydrothermal method with electrospinning as the template and carbon source for the first time [155]. The doping of carbon can reduce part of Co^3+^ to Co^2+^, which provides more Co^2+^ and produces oxygen vacancies. Co^2+^ has higher electronegativity, which improves the capacity and enhances the rate capability of materials. At the same time, the introduction of oxygen vacancies and the hollow structure of the material shorten the Li^+^ diffusion path and provide enough voids to accommodate the volume change. In addition, density functional theory (DFT) calculation further explains the conductivity enhancement of C-doped Co_3_O_4_ hollow nanofibers, which is related to the delocalization of its larger energy band structure, and further proves that this method can be used to prepare other transition metal oxide anodes. Compared with undoped Co_3_O_4_ hollow nanofibers (initial discharge and charge capacity of 1230 and 849 mA h g^−1^, respectively, the initial discharge and charge capacity of C-doped Co_3_O_4_ increased to 1385 and 972 mA h g^−1^. At the same time, after 100 charge-discharge cycles, the reversible capacity of carbon-doped Co_3_O_4_ remained at 1121 mA h g^−1^, which was higher than that of undoped Co_3_O_4_ (663 mA h g^−1^). Although the electrochemical properties of Co_3_O_4_ hollow nanofibers have been improved after doping with carbon, with the continuous exploration of scholars in recent years, some other doping methods have appeared, one after another. For example, in 2015, Wang et al. successfully controlled the diameter of Co_3_O_4_ particles to about 5 nm, and synthesized a Co_3_O_4_-carbon nanosheet structure in which nano-Co_3_O_4_ was anchored in the three-dimensional array of carbon nanosheets [156]. When used as anode material for lithium-ion batteries, its reversible specific volume can exceed 1200 mA h g^−1^. Yang and his colleagues successfully prepared Co_3_O_4_/onion-like carbon (OLC) nanocomposites with high area by a simple solvent method [157]. Nanoparticles of onion carbon composed of quasi-spherical graphite shell have isotropic surface, and lithium-ions can pass through electrolyte from all directions, which increases the conduction rate of lithium-ions. When used as anode material, the specific capacity was kept at 632 mA h g^−1^ after 100 cycles at the current density of 200 mA h g^−1^, and the anode material showed good electrochemical performance, in which the onion-like structure played an important role. Similarly, Wang et al. successfully synthesized Co_3_O_4_/carbon (Co_3_O_4_/C) materials with onion-like structure by carbothermal reduction and low-temperature oxidation with (BMIM)N(CN)_2_ as precursor, in which Co_3_O_4_ was embedded in carbon matrix and showed good performance as anode material for lithium-ion batteries [158].

In addition, the introduction of porous carbon into Co_3_O_4_ also shows good effect. In the research of Han et al., Co_3_O_4_ nanoparticles (NPs) were uniformly immobilized on porous carbon (PC) by solvothermal method and simple heat treatment process, and cobalt oxide modified porous carbon was prepared [159]. The three-dimensional layered porous structure of the composites and the small size of Co_3_O_4_ NPs promote the diffusion of electrolytes and the insertion of lithium ions. Moreover, the interconnection network of porous carbon not only reduces the internal resistance of lithium-ion batteries, but also provides a mechanical buffer to prevent large volume changes, which plays a role in maintaining structural stability. When used as anode material for lithium-ion batteries, the composite material shows excellent cycle performance, high conductivity, and good rate performance. Co_3_O_4_/PC composites can make full use of the advantages of Co_3_O_4_ and hierarchical porous carbons and show reversible properties of 654 mA h g^−1^. It is proven that the composite of porous carbon materials and cobalt oxide materials is an effective method to improve the electrochemical properties of single cobalt oxide materials. In addition, the effectiveness of this method is also verified in the research of cobalt oxide compounding with porous carbon materials derived from carbon dioxide [19]. The synthesized cobalt oxide/porous carbon composites have a reversible capacity of 1179 mA h g^−1^ (higher than Co_3_O_4_/PC), and the reversible capacity remains at 1179 mA h g^−1^ after 300 cycles at 1000 mA g^−1^.

Gu et al. prepared a new type of active porous carbon (APC) with lotus-like layered porous structure, and fixed cobalt oxide nanoparticles on the surface and pores of APC, to successfully prepare Co_3_O_4_/APC nanocomposites [20]. APC, which has a large number of mesoporous and microporous structures, provides more active sites for the loading of nano-Co_3_O_4_ particles, reduces the agglomeration of Co_3_O_4_ particles on the surface, and can effectively alleviate the volume expansion effect. When the composite material is used as anode material for the lithium-ion battery, the reversible capacity is 625 mA h g^−1^ after 200 cycles at the current density of 500 mA g^−1^, which is better than pure Co_3_O_4_ and Co_3_O_4_/PC composites. In addition, the displayed discharge capacity at a current density of 1000 mA g^−1^ is about 360 mA g^−1^, and the reversible capacity becomes 810 mA g^−1^ when the current density drops to 100 mA g^−1^ again. Among them, APC is prepared by activation and carbonization with waste loose manure as carbon source. The synthesis method is simple, the raw material cost is low, and it can be prepared in large quantities, which provides a new method for the preparation of cobalt oxide-carbon nanocomposites.

To sum up, the composite of carbon and Co_3_O_4_ nanomaterials plays a better role in improving the electrochemical properties of Co_3_O_4_ materials and has become one of the main methods to improve the electrochemical properties of Co_3_O_4_ materials, which has been studied more. Comparatively speaking, carbon materials with porous structure have a stronger effect on enhancing the electrochemical performance of Co_3_O_4_ materials than ordinary carbon materials, which is related to its porous structure. In future research, we can further explore ways to improve the contact area between electrodes and electrolytes and increase the number of holes.

### 4.2. Co Sulfide-Carbon Composite in LIBs Anode 

It has been proven that cobalt sulfide has better electrochemical performance than cobalt oxide, and the introduction of carbon nanomaterials into cobalt oxide has a very good effect. Therefore, researchers have also explored many methods to introduce carbon nanomaterials into cobalt sulfide and achieved good results.

Similar to cobalt oxide mentioned earlier, porous carbon materials have attracted more attention in the composite research of cobalt sulfide and carbon nanomaterials. The cobalt sulfide nanocomposite with porous carbon fiber intercalated (Co_9_S_8_@C), prepared by simple carbonization and sulfurization steps, shows good performance as an anode material for lithium ion batteries, which proves that the composite of porous carbon and cobalt sulfide is a method to improve its electrochemical performance [160]. The porous structure of carbon and the mesoporous structure of Co_9_S_8_ enlarge the interface between electrode and electrolyte, promote the transport of lithium ions, and prevent the aggregation and volume change of Co_9_S_8_ particles, thus improving the electrochemical performance. Its reversible capacity can reach 1565 mA h g^−1^ at 0.1 C rate. In the aspect of cycle stability, after 100 cycles, the specific discharge capacity remained at 872 mA h g^−1^ (Co_9_S_8_ was 294 mA h g^−1^), while after 300 cycles, it still had a reversible capacity of about 606 mA h g^−^^1^. Huang and his colleagues prepared polycrystalline Co_9_S_8_ coated with porous carbon [161]. Because of its porous structure and carbon coating, it shows excellent electrochemical performance as an anode material for lithium-ion batteries. After 400 cycles at 2 A g^−1^, the capacity retention rate is 84.7%, and it has great prospects as a LIB anode material.

In addition, the composite of carbon nanosheets, carbon nanotubes, and cobalt sulfide also shows good results and has been studied by a large number of scholars. In the research of Shi et al., cobalt sulfide/carbon composites were prepared from metal-organic framework, and the flaky three-dimensional hollow cobalt sulfide/carbon nanosheet array (h-Co_4_S_3_/CNA@CC) grown on carbon cloth has good rate ability and cycle stability when used as anode of lithium-ion battery [162]. Among them, Co_4_S_3_ is wrapped in graphite carbon cage like pomegranate, and this unique structure can effectively buffer the volume expansion in physical and chemical process. In addition, carbon cloth is used as the substrate, which provides a large number of active sites and avoids the use of additives. As the anode of a lithium-ion battery, it has 654.3 mA h g^−1^ and 394.1 mA h g^−1^ at current densities of 1 A g^−1^ and 2 A g^−1^, respectively. After 500 cycles at a current density of 2 A g^−1^, the capacity retention rate reaches 79%.

In addition, Wang et al. successfully synthesized layered CoS_1-x_ nanoparticles/porous carbon nanoparticles (CoS_1−x_@PCSs) by annealing and vulcanization with Co(OH)_2_ nanoparticles and sodium citrate as precursors. CoS_1−x_@PCSs composites have a large number of active sites and abundant nano-pores on the surface of carbon sheets, which can effectively improve the ion/electron transfer efficiency [163]. At the same time, the thin carbon structure encapsulates CoS_1-x_ nanoparticles, which can prevent cobalt sulfide from aggregating in the electrochemical process, buffer the volume change, isolate active substances and electrolytes, and improve the conductivity. The synergistic effect between the cobalt sulfide and carbon coating makes CoS_1−x_@PCSs composites have excellent lithium storage performance. As an anode of lithium-ion battery, it has a reversible capacity of 1199.6 mA h g^−1^ after 100 cycles at 0.1 A g^−1^, and a capacity of 825.7 mA h g^−1^ after 800 cycles at a current density of 1 A g^−1^. In addition, in the research of Wu and his colleagues, a porous carbon polyhedron/carbon nanotube hybrid was combined with cobalt sulfide material to synthesize three-dimensional hollow cobalt sulfide@porous carbon polyhedron/carbon nanotube hybrid (CoS_1−x_@PCP/CNTs) material [164]. As shown in Figure 15a,b, the product obtained at 600 °C has a discharge capacity of 1668 mA h g^−1^ at a current density of 0.2 A g^−1^ after 100 cycles, and its cycle performance is excellent, which is higher than that of the product obtained at 500 and 700 °C. With its excellent cycle performance, its charge and discharge capacities in the first cycle at 0.2 A g^−1^ are 1246 and 2083 mA h g^−1^, respectively, with an irreversible capacity loss of about 40%.

As shown in Table 2, cobalt-carbon composite nanomaterials show excellent electrochemical performance compared with cobalt compounds and carbon nanomaterials alone. To sum up, combining cobalt-containing nanomaterials with carbon nanomaterials and controlling the micro-morphology of the composites (such as improving the porosity of the materials and constructing porous network structure, etc.) to make them have a larger contact area with electrolytes is an effective method to improve the electrochemical performance of materials, which has great application potential in the field of anode materials for lithium-ion batteries.

## 5. Summary and Outlook

Generally speaking, this paper summarizes the progress and challenges of the application of cobalt nanomaterials, carbon nanomaterials, and their composites in lithium-ion batteries. In the research of lithium batteries, the Co_3_O_4_ anode has excellent performance, which can significantly improve the performance of LIBs. However, the shortcomings of fast capacity attenuation and low rates of performance limit its practical application. The difference is that binary cobalt-containing oxides and cobalt-containing sulfides have higher specific capacitance than Co_3_O_4_, and their volume changes during charge and discharge can be reduced by adjusting their micro-morphology or compounding with other materials, thus improving their electrochemical cycle performance. As the traditional anode materials in electrochemical energy storage devices, carbon nanotubes, carbon nanosheets, carbon nanospheres, graphene, and other carbonaceous materials still occupy an irreplaceable position in the application of lithium-ion batteries. At present, the composite of cobalt-containing nanomaterials and carbon materials is one of the main means to improve the electrochemical performance of cobalt nanomaterials. The synergistic effect between them can effectively improve the properties of composites, which have excellent electrochemical performance as anodes of lithium-ion batteries.

Nano-sized cobalt-containing compounds have rich natural reserves, strong electrical conductivity, and large theoretical specific capacitance. In the past twenty years, they have been widely used and developed. Although there are some shortcomings in practical application, such as low energy density and short cycle life, their electrochemical performance has been significantly improved by various methods. Moreover, with the continuous improvement of researchers’ proficiency with regard to the production process, nano-cobalt-containing compounds have been used in commercial fields. Carbon nanomaterials also show excellent performance as anode materials for lithium-ion batteries. In the future, it will be necessary to further explore methods to accurately control their morphology and structure. At the same time, while ensuring their electrochemical performance, the preparation process should be simplified, and the cost should be reduced as much as possible.

It should also be noted that although cobalt resources are abundant at present, some scholars have predicted that they may not be able to meet demand by around 2050. Therefore, in order to ensure the further research and the sustainable utilization of cobalt-containing nanomaterials, it is of great significance to add other metals appropriately to reduce the usage of cobalt while ensuring the performance of batteries, and to further develop the recycling and utilization technology of lithium-ion batteries.

## Figures and Tables

**Figure 1 nanomaterials-12-02042-f001:**
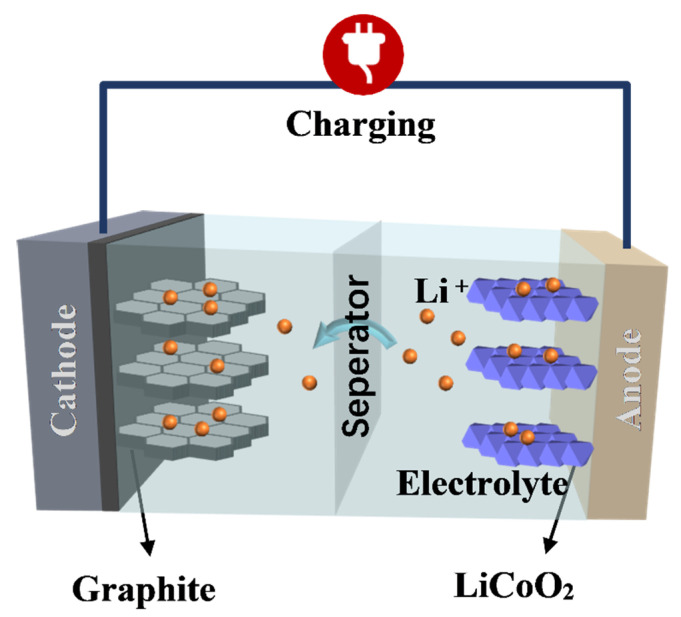
Schematic diagram of lithium-ion battery charging process.

**Figure 2 nanomaterials-12-02042-f002:**
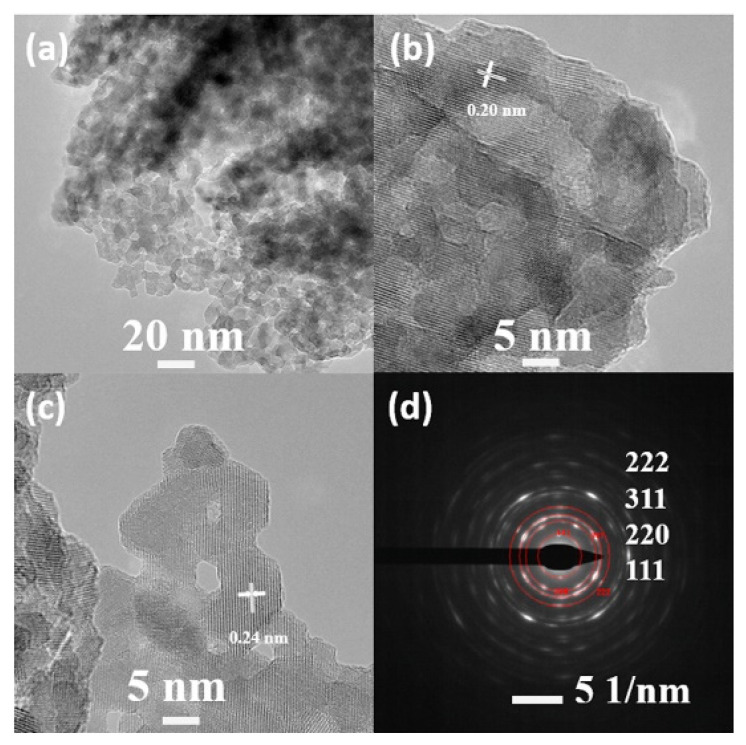
(**a**–**d**) TEM and HR-TEM images of a MnCo_2_O_4_ microsphere. Reproduced with permission from Mobinul Islam, A high voltage Li-ion full-cell battery with MnCo_2_O_4_/LiCoPO_4_ electrodes; published by Elsevier, 2020 [12].

**Figure 3 nanomaterials-12-02042-f003:**
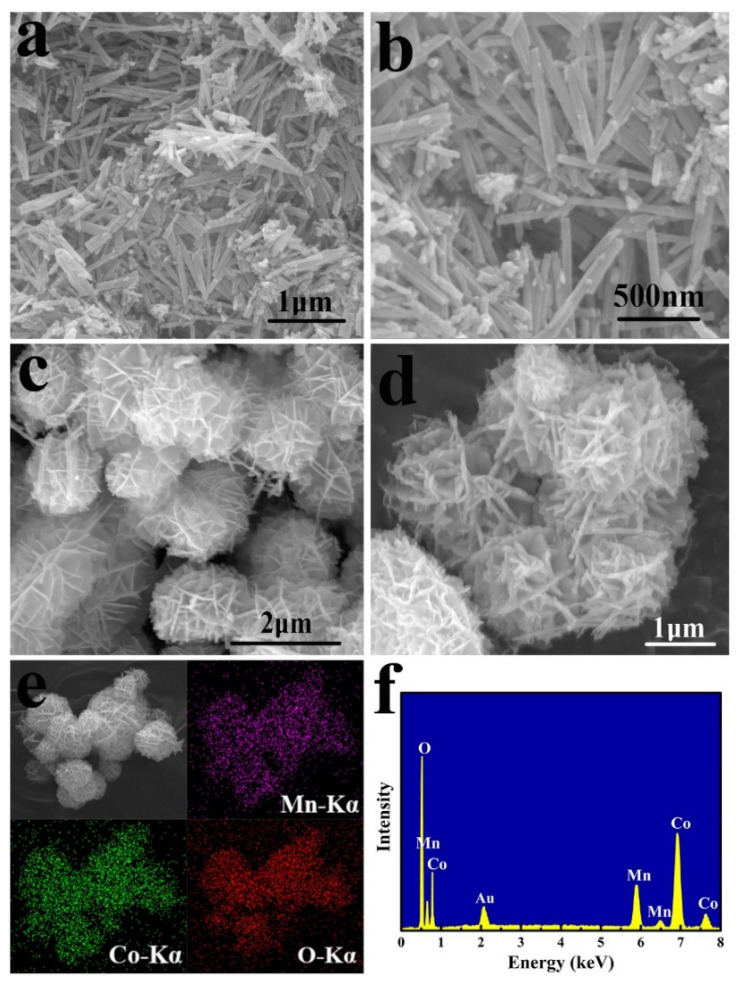
(**a**,**b**) The FE-SEM images of r-MCO; (**c**,**d**) The FE-SEM images of h-MCO; (**e**) The element mappings of h-MCO; (**f**) The EDS of h-MCO. Reproduced with permission from Xiaolan Song, Hydrothermal synthesis of porous hydrangea-like MnCo_2_O_4_ as anode materials for high performance lithium-ion batteries; published by Elsevier, 2019 [58].

**Figure 4 nanomaterials-12-02042-f004:**
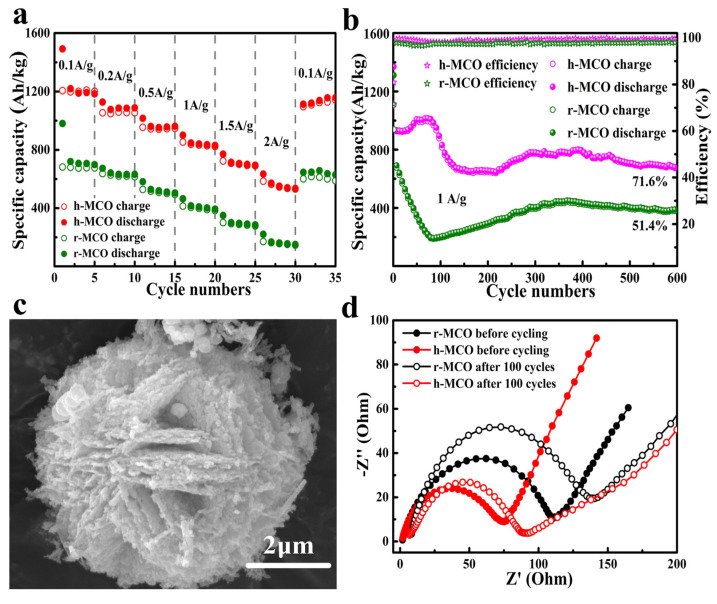
(**a**) Rate performances of the MCOs; (**b**) Long cycling performances of the MCOs at 1 A g^−1^; (**c**) FE-SEM image of h-MCO after 5 cycles at 0.1 A g^−1^; (**d**) Nyquist profiles of MCOs before cycling and after 100 cycles. Reproduced with permission from Xiaolan Song, Hydrothermal synthesis of porous hydrangea-like MnCo_2_O_4_ as anode materials for high performance lithium-ion batteries; published by Elsevier, 2019 [58].

**Figure 5 nanomaterials-12-02042-f005:**
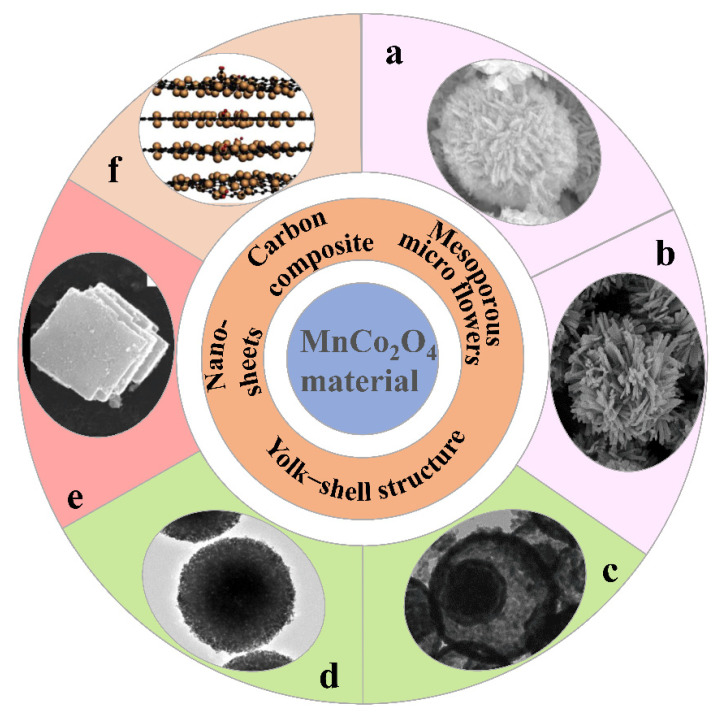
(**a**,**b**) Mesoporous micro flowers; Reprinted with permission from ref. [48]. Copyright 2016, Elsevier. Reprinted with permission from ref. [61]. Copyright 2015, Materials Letters. (**c**,**d**) Yolk-shell structure; Reprinted with permission from ref. [62]. Copyright 2013, ACS Applied Materials & Interfaces. Reprinted with permission from ref. [51]. Copyright 2017, Journal of Alloys and Compounds. (**e**) Nanosheets; Reprinted with permission from ref. [54]. Copyright 2015, Elsevier. (**f**) Carbon composite; Reprinted with permission from ref. [50]. Copyright 2016, Elsevier.

**Figure 6 nanomaterials-12-02042-f006:**
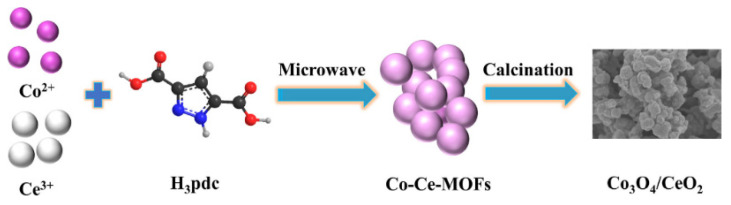
Schematic illustration of Co_3_O_4_/CeO_2_ preparation. Reproduced with permission from Ying Kang, Highly efficient Co_3_O_4_/CeO_2_ heterostructure as anode for lithium-ion batteries; published by Elsevier, 2020 [11].

**Figure 7 nanomaterials-12-02042-f007:**
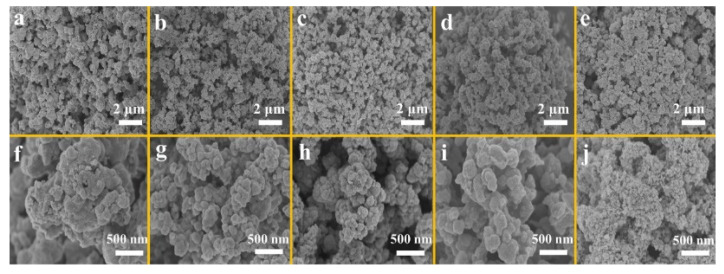
SEM images of (**a**,**f**) Co_3_O_4_, (**b**,**g**) Co_3_O_4_/CeO_2_, (**c**,**h**) 3Co_3_O_4_/CeO_2_ (Co_3_O_4_/CeO_2_ having the molar ratio of Co/Ce = 3:1), (**d**,**i**) 5Co_3_O_4_/CeO_2_ (Co_3_O_4_/CeO_2_ having the molar ratio of Co/Ce = 5:1), and (**e**,**j**) 7Co_3_O_4_/CeO_2_ (Co_3_O_4_/CeO_2_ having the molar ratio of Co/Ce = 7:1). Reproduced with permission from Ying Kang, Highly efficient Co_3_O_4_/CeO_2_ heterostructure as anode for lithium-ion batteries; published by Elsevier, 2020 [11].

**Figure 8 nanomaterials-12-02042-f008:**
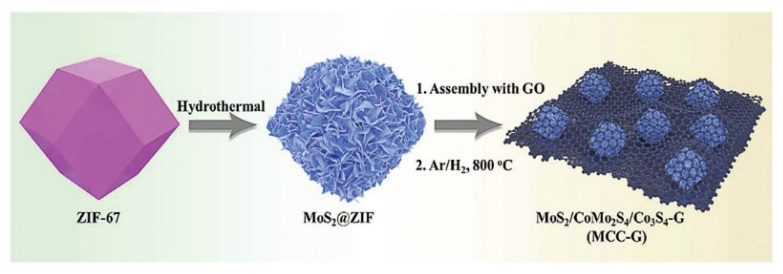
Schematic illustration for the fabrication of MoS_2_/CoMo_2_S_4_/Co_3_S_4_ supported by graphene. Reproduced with permission from Pengcheng Wang, Constructing MoS_2_/CoMo_2_S_4_/Co_3_S_4_ nanostructures supported by graphene layers as the anode for lithium-ion batteries; published by Royal Society of Chemistry, 2020 [71].

**Figure 9 nanomaterials-12-02042-f009:**
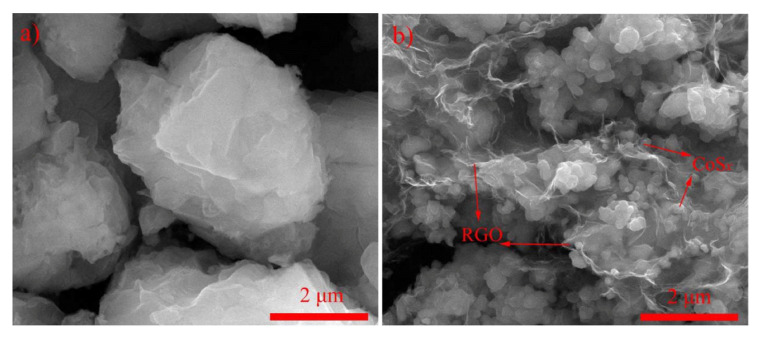
SEM images of (**a**) CoS_x_; (**b**) CoS_x_/RGO. Reproduced with permission from Junsheng Zhu, Embedding cobalt sulfide in reduced graphene oxide for superior lithium-ion storage; published by Elsevier, 2019 [82].

**Figure 10 nanomaterials-12-02042-f010:**
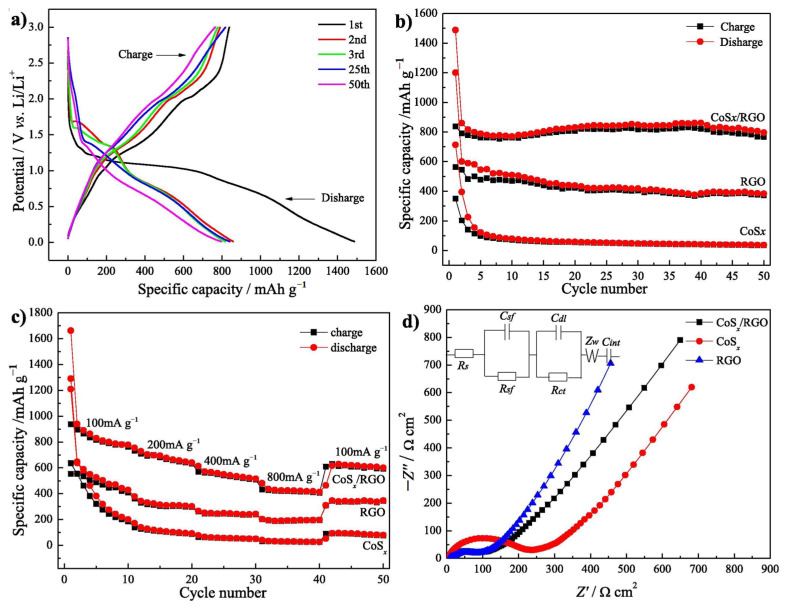
(**a**) GCD curves of CoS_x_/RGO; (**b**) Cycling performance of the samples; (**c**) Rate capability and (**d**) Nyquist plots of CoS_x_, RGO and CoS_x_/RGO. Reproduced with permission from Junsheng Zhu, Embedding cobalt sulfide in reduced graphene oxide for superior lithium-ion storage; published by Elsevier, 2019 [82].

**Figure 11 nanomaterials-12-02042-f011:**
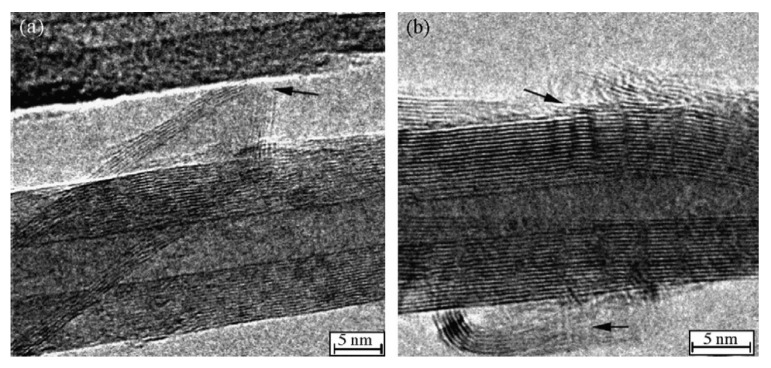
HRTEM images of etched CNTs. The black arrows in figures represent the etched position. Reproduced with permission from Shubin Yang, Electrochemical performance of arc-produced CNTs as anode material for lithium-ion batteries; published by Elsevier, 2007 [93].

**Figure 12 nanomaterials-12-02042-f012:**
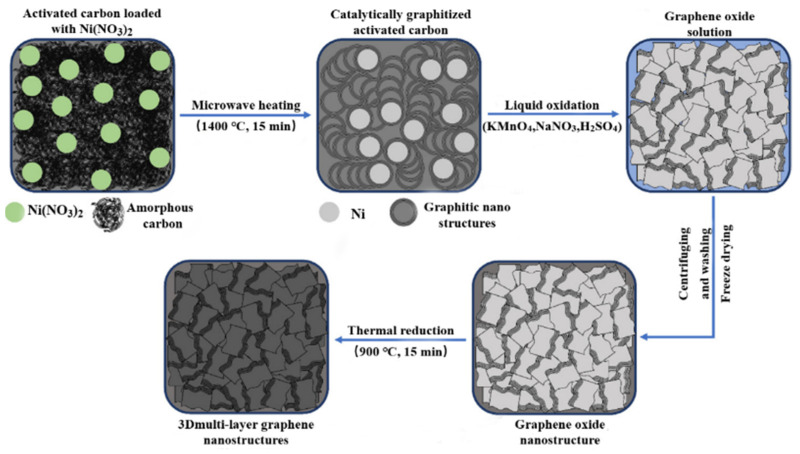
Schematic diagram of the synthesis process of 3DMGs. Reproduced with permission from SalmanKhoshk Rish, Novel composite nano-materials with 3D multilayer-graphene structures from biomass-based activated-carbon for ultrahigh Li-ion battery performance; published by Elsevier, 2021 [111].

**Figure 13 nanomaterials-12-02042-f013:**
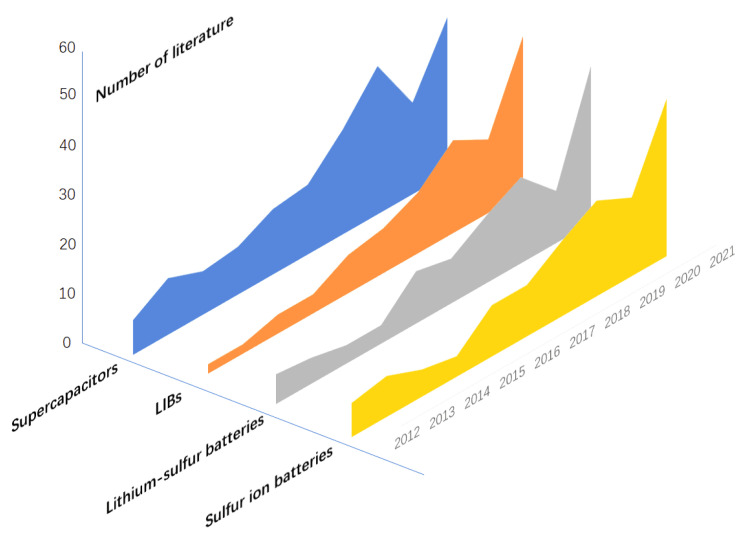
Research situation of CSs in LIBs, Lithium-sulfur batteries and Sulfur ion batteries from 2012 to 2021.

**Figure 14 nanomaterials-12-02042-f014:**
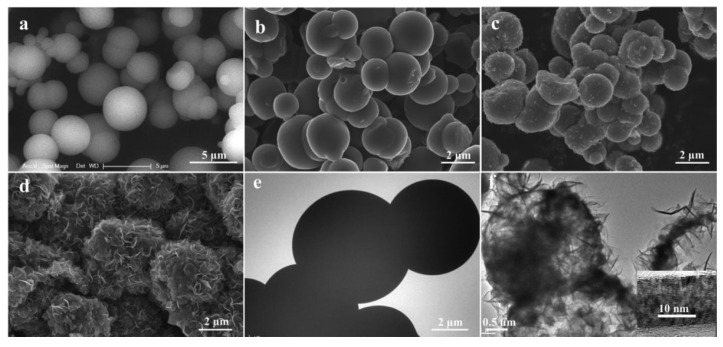
(**a**–**d**) SEM images of AMC before (**a**,**b**) And after (**c**,**d**) The electrochemical graphitization at 820 °C. (**a**) Raw; (**b**) Immersed in the melt for 2 h; (**c**) 0.45 V for 0.25 h; (**d**) 0.45 V for 2 h; (**e**,**f**) TEM images of AMC; (**e**) And the graphitized AMC at 820 °C and 0.45 V for 2 h. Reproduced with permission from Chunyan Zhang, Glucose-derived hollow microsphere graphite with a nanosheets-constructed porous shell for improved lithium storage; published by Elsevier, 2021 [131].

**Figure 15 nanomaterials-12-02042-f015:**
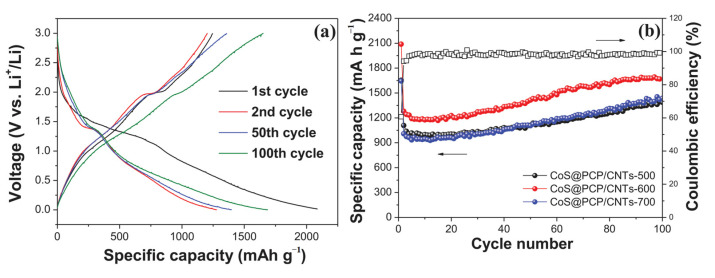
(**a**) Galvanostatic charge-discharge voltage profiles of 3D hollow CoS@PCP/CNTs-600 polyhedral at the current density of 0.2 A g^−1^; (**b**) Cycling performances of CoS@PCP/CNTs-500, CoS@PCP/CNTs-600, and CoS@PCP/CNTs-700 and Coulombic efficiency of CoS@PCP/CNTs-600 at the current density of 0.2 A g^−1^. Reproduced with permission from Wu, In-Situ Formation of Hollow Hybrids Composed of Cobalt Sulfides Embedded within Porous Carbon Polyhedra/Carbon Nanotubes for High-Performance Lithium-Ion Batteries; published by Advanced Materials, 2015 [164].

**Table 1 nanomaterials-12-02042-t001:** Comparison of electrochemical properties of carbon microspheres, graphene and carbon nanotubes.

Types of Materials	Cycle Number	Current Density	Reversible Capacity (mA h g^−1^)	Structure	Reference
Nanoporous carbon microspheres	500	200 mA g^−1^	650	double-carbon-shell	[130]
NCMs	50	210 mA g^−1^	926.654	porous spherical	[121]
AMC	100	1860 mA g^−1^	280	hollow microsphere	[131]
GNSs	30	0.2 mA cm^−2^	502	disordered graphene layers	[109]
N-GNSs	100	100 mA g^−1^	760	porous architecture	[139]
Porous graphene film	50	50 mA g^−1^	195	porous structure	[140]
3DMGS	170	100 mA g^−1^	1513.2	multilayer nanostructures	[111]
CNTs	30	0.2 mA cm^−2^	266	single-walled	[96]
VA-CNTs	50	100 mA g^−1^	350	multi-walled	[97]

**Table 2 nanomaterials-12-02042-t002:** Comparison of electrochemical properties of cobalt compounds, carbon nanomaterials and cobalt-carbon composites.

Types of Materials	Current Density (A g^−1^)	Structure	Cycle Number	Reversible Capacity (mA h g^−1^)	Reference
Co_3_O_4_	1.8 (2C)	porous disk-like	100	304	[165]
Ultrathin mesoporous Co_3_O_4_ nanosheet	2	irregular small blocks	50	479	[166]
Co_3_O_4_	0.4	porous microspheres	40	654	[167]
Co_3_O_4_	0.8	mesoporous microstructure	50	260	[168]
Co_3_O_4_	2	porous nanorod	25	516	[169]
Co_3_O_4_	2	nanocages	25	252	[38]
Co_3_O_4_ microspheres	0.1	hollow and porous	520	550	[40]
Co_3_O_4_ Nanosheet Arrays	0.1	mesoporous	80	1576.9	[41]
CoS_2_	0.1	hollow spheres	40	320	[170]
2D Co_3_S_4_	0.7	nano thickness sheet-like	400	416	[171]
MnCo_2_O_4_	0.4	core-shell ellipsoidal	150	620.0	[53]
MnCo_2_O_4_	0.1	nano sheet	50	681	[54]
MnCo_2_O_4_	1	hierarchical porous	1000	740	[53]
CoMn_2_O_4_	1	microspheres	1000	420	[57]
MnCo_2_O_4_	0.1	porous hydrangea-like	100	930	[58]
Co_9_S_8_	2	hollow nanospheres of mesoporous	800	896	[73]
Nitrogen-Doped Carbon Nanosheets	1	porous nanosheets	1000	222	[172]
Carbon	0.05	hierarchical porous	50	877.9	[173]
Porous graphene film	10	porous structure	10,000	971	[140]
3D multilayer-graphene	5	cloud-like	1100	260.3	[111]
N-doped carbon nanospheres	0.1	nanospheres	150	505	[174]
N-doped graphene	0.1	nanosheets	100	452	[175]
Hierarchical porous Carbon microspheres	1	hierarchical porous microspheres	70	200	[127]
C-Co_9_S_8_	2	hierarchical porous	400	476	[161]
MnCo_2_O_4_/C	0.05	quasi-hollow microspheres	50	488	[56]
Co_3_O_4_/APC (Activated porous carbon)	0.5	lotus root-like and layered porous	200	625	[20]
Co/CPC (CO_2_-derived porous carbon)	1	porous	300	1179	[19]
Co_3_O_4_@CNT	0.01	mesoporous peapod-like	100	700	[176]
Co_3_O_4_@Graphene core-shell	1	mesoporous hollow spheres	200	700	[18]
CoO-Co_3_O_4_-RGO	0.1	sandwich-like	200	994	[177]
Carbon@cobaltous oxide	0.2	nanofiber	100	892	[17]
CoP@C	0.5	uniform spherical morphology	1000	483.4	[178]
Co_1-X_S-CNFs	2	onion-like carbonaceous	500	252	[179]

## Data Availability

Not applicable.

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
