# Peer review of "A Review of Cobalt-Containing Nanomaterials, Carbon Nanomaterials and Their Composites in Preparation Methods and Application"

_nanomaterials, 2022, doi:10.3390/nano12122042_

Round 1
Reviewer 1 Report
Reviewer’s comments
Manuscript Number: nanomaterials-1737988
Title: Application of cobalt and carbon nano materials in anode mate-rials of LIBs
Journal: Nanomaterials
- The abstract is very general, the main findings should be specified and highlighted.
- Why cobalt and carbon only, since there are so many materials used for LiBs. The reasons should be clearly mentioned in the introduction.
- Although it is a review article, some new figures, diagrams, or designs should be provided with the copied figures to show the authors' input.
- Although there is a section on “Supplementary Materials: The following supporting information can be downloaded at: 678 www.mdpi.com/xxx/s1, Figure S1: title; Table S1: title; Video S1: title.” The Supp. Mater. file is not available on the system.
- The abbreviation should be used in a standard way, where there are forms are used for lithium ion batteries (LiBs, Libs, Lib, LIB).
- The reference number should be added to the figure caption.
- The discussion is very simple and surface. A deep discussion is needed.
- The correlation between electrochemical predominance and morphological characteristics should be discussed. Here is some literature that may help: Phys. Chem. C 122, 12200-12206 (2018) 10.1021/acs.jpcc.8b03306, Journal of Solid State Electrochemistry 18, 2505-2512 (2014) 10.1007/s10008-014-2510-3. Journal of Materials Science 54, 683-692 (2018) 10.1007/s10853-018-2871-6
- Detailed tables of comparison should be added to summarise and compare the performance of cobalt and carbon materials for LiBs.
Reviewer 2 Report
I have read the review article titled “Application of cobalt and carbon nanomaterials in anode materials of LIBs” by Haiwang Wang et al submitted to Nanomaterials. The review itself is quite comprehensive and well-written while encompassing both the cathode and anode materials for battery application on cobalt oxides as potential electrodes.
The demand for advanced electrochemical energy storage devices with increased power and energy densities is increasing due to sustainable energy and environmental issues. Hence, electrical energy storage systems such as batteries and supercapacitors will play a significant role. Therefore, this extensive review article on a cobalt oxide/sulfide structure is significant. Though electrodes for batteries is a well-researched area but review article focussing on Co as an alternative to Carbon with improved performance and unique morphology could be new and the review content is useful to the energy community. The review is fairly well written with neat characterizations adopted from the other articles. However, revision is required to clarify some parts of the review, before rendering a final decision.
Some parts of the submitted review must be improved.
Following are my general and specific comments:
- Title can be a bit more effective. Intended meaning of the review is unclear.
- Section 1; line 27 “people” are not interested but researchers and energy community is interested. General public is not interested in energy storage research but energy community has interest.
- Line 37-38 Carbon has EDLC behaviour.
- Line 39; TM compounds can store more energy but less cyclability than carbon.
- Section 2; paragraph 2, LiCoO2 has been used as an anode in LIBs as invented by SONY. However, this material has been proposed for the anode in this review, Justify.
- A relevant work on this Cobalt material reported in the literature needs to be discussed (doi.org/10.1016/j.progsolidstchem.2020.100298; doi.org/10.3390/nano7110356) and the role of morphology, and dopants.
Specific
- Section 2.1 – lines 81 -82 need to be rewritten.
- Their energy density 83 is limited [8]” -please provide the value.
- Lines 87 – 88 – an example of a high-capacity anode should be given.
- Lines 91- 92 – binary transition metal oxides, phosphates, and molybdates are also widely used as anodes such as (NiMoO4, CaMoO4 MgMoO4, LiCo/NiPO4, etc.) for LIBs and SIBs, Manickam Minakshi et al has published several articles in this area. These are missing.
- Line 117 – Islama et al – no reference.
- Line 120 – LCOPO4 should read as LiCoPO4.
- Line 116 – Xu et al – no reference.
- The equations on Page 4 are given but the reactions are not explained.
- Figures 3c-d are not explained.
- Line 176 “The micromorphology is analyzed, as shown in Figure 22” is it Figure 22?
- Line 263 – “low specific capacity” how much, provide the value, please.
- Sections, 3, 3.1 - 2 title “LIB” should be in capital letters.
- How carbon nanotubes and graphenes compare among themselves in terms of the anode in LIBs – please provide specific capacity and longevity values.
- How the micro-morphology can be controlled to have a larger contact area?
- Maybe a good idea to provide a table comparing carbon microspheres, graphene, carbon nanotube, etc for their electrochemical performance.
Round 2
Reviewer 1 Report
The authors have addressed most of the comments in the revised version. The current version could be accepted for publication.
Author Response
Dear reviewer,
On behalf of my co-authors, we thank you very much for your affirmation and support of this article, and we appreciate you very much for you positive and constructive comments and suggestions on our manuscript entitled “A review of cobalt-containing nanomaterials, carbon nanomaterials and their composites in preparation methods and application (ID: nanomaterials-1737988)”.
Thank you and best regards.
Yours sincerely,
Haiwang Wang
Corresponding author:
Name: Haiwang Wang
E-mail: whwdbdx@126.com
Reviewer 2 Report
I have read the revised version of the manuscript (having a revised title namely (A review of cobalt-containing nanomaterials, carbon nanomaterials, and their composites in preparation methods and application). The revised title and the scientific content have some merits, but the revised part of the manuscript is dreadful.
I am sorry to say that the authors rushed and submitted the revised manuscript without paying close attention. The authors must understand the impact factor (IF) of this journal is around 5 and the current submission deserves no merit. It has been poorly drafted. The authors need to take every opportunity to polish their work, which improves the quality. However, this is not the case seen here.
The following are just a few examples of how poorly the authors have revised the manuscript with no facts.
· Figure 1 caption - stated as an "electrolysis" process.
Authors need to understand fundamental electrochemistry. The galvanic cell (battery) produces an electric current in which a spontaneous chemical reaction is used to generate electrical energy. While electrolysis is the opposite process. Remove the word "electrolysis"
· Page 4, line 134 expresses energy density in W h L-1. This is completely unacceptable.
The energy density should be expressed in mAh/g while the volumetric energy density is Wh/L
· "Delirium States"??
· Page 4, line 138: how the structural instability of LCO could result in a safety problem.
Please make the above sentence meaningful. It is scientifically incorrect.
· Line 167 - Why redox is in the capital letter? (REDOX)
· Line 183; what is ACPKS?
· Line 189 - 193: "In general, the electrochemical performance can be significantly improved by compounding cobalt compounds with other materials, among which the compounding of cobalt compounds with carbon nanomaterials is more studied. Therefore, the application of cobalt compounds, carbon nanomaterials and their composites in lithium-ion batteries will be discussed in this paper"
The above paragraph is awkwardly written.
· Line 235: What is LDH, NSA, T2NSA?
· The reaction mechanism explained in lines 288 - 292 is OK but requires more attention in polishing the language and being scientifically correct while addressing all the equations appropriately.
· Figure 5 caption is dreadful. It reads as though a paragraph.
· All the scientific units are poorly drafted and in some places they are incorrect.
Author Response
Dear reviewer,
On behalf of my co-authors, we thank you very much for giving us an opportunity to revise our manuscript again, and we appreciate you very much for your positive and constructive comments and suggestions on our manuscript entitled “A review of cobalt-containing nanomaterials, carbon nanomaterials and their composites in preparation methods and application (ID: nanomaterials-1737988)”.
We have studied your comments carefully and made changes to the article by using the "Track Changes" function. We have tried our best to revise our manuscript according to the comments.
We would like to express our great appreciation to you for comments on our paper. Looking forward to hearing from you.
Thank you and best regards.
Yours sincerely,
Haiwang Wang
Corresponding author:
Name: Haiwang Wang
E-mail: whwdbdx@126.com
Round 3
Reviewer 2 Report
The authors have given fairly careful consideration, and the revised review article reads OK.
This manuscript is a resubmission of an earlier submission. The following is a list of the peer review reports and author responses from that submission.